# Mapping the AMR Infection Landscape in Bihar: Implications for Strengthening Policy and Clinical Practice

**DOI:** 10.3390/antibiotics14070684

**Published:** 2025-07-05

**Authors:** Vinay Modgil, Sundeep Sahay, Neelam Taneja, Burhanuddin Qayyumi, Ravikant Singh, Arunima Mukherjee, Bibekananda Bhoi, Gitika Arora

**Affiliations:** 1Society for Health Information Systems Programmes (HISP India), New Delhi 110025, India; vinay.modgil@hispindia.org (V.M.); arunimam@ifi.uio.no (A.M.); gitika.arora@hispindia.org (G.A.); 2Department of Informatics, University of Oslo, 0373 Oslo, Norway; 3Department of Medical Microbiology, Postgraduate Institute of Medical Education and Research (PGIMER), Chandigarh 160012, India; taneja.neelam@pgimer.edu.in; 4Homi Bhabha Cancer Hospital and Research Centre, Muzaffarpur 842004, India; qayyumibn@hbchrcmzp.tmc.gov.in (B.Q.); ravikant@hbchrcmzp.tmc.gov.in (R.S.); bhoibibek@gmail.com (B.B.); 5Institute of Health and Society (HELSAM), University of Oslo, 0316 Oslo, Norway

**Keywords:** antimicrobial resistance, digital health, surveillance systems, clinical microbiology, antimicrobial stewardship

## Abstract

**Background:** Antimicrobial resistance (AMR) poses a significant threat to public health, especially in low- and middle-income countries (LMICs), where surveillance infrastructure is underdeveloped. Bihar, India’s third most populous state and one of its least-resourced states, has remained largely absent from national AMR monitoring initiatives. **Methods:** This study aimed to characterize the AMR infection landscape across five public tertiary care hospitals in Bihar over three years (2022–2024) and to assess the feasibility of integrating digital workflows for real-time microbiological reporting. Standardized antimicrobial susceptibility testing (AST) was performed on >48,000 urine, pus, and blood samples using CLSI guidelines. Facility-level data were digitized into an open-source AMR reporting system, enabling automated antibiogram generation. **Results:** The findings revealed substantial resistance: high resistance to beta-lactams, carbapenems, and fluoroquinolones across pathogens. For instance, *E. coli* sensitivity to nitrofurantoin varied from 86.5% at NMCH (Patna) to 44.7% at JLNMCH (Bhagalpur), while cephalosporin sensitivity in *Klebsiella* spp. dropped below 2% in several hospitals. MRSA prevalence exceeded 65% in two facilities, far above the national average of 47.8%. Digital integration led to a four-fold increase in culture testing in all facilities and improved data completeness and turnaround times. Spatial analysis and microbiology laboratory assessment revealed significant geographic disparities in diagnostic access, with facilities in remote districts facing delays of over four hours for basic testing. **Conclusions:** Our study is the first study from India to create such a broad, facility-associated AMR picture over time at a state level. Policy implications include the need for a state-level AMR surveillance dashboard, alignment of procurement with facility-specific resistance patterns, and routine stewardship audits. Clinically, this study demonstrates the utility of localized antibiograms for guiding empirical therapy in resource-limited settings. This study provides a scalable framework for embedding AMR surveillance into routine health system workflows in LMICs.

## 1. Introduction

A vital source of information required to navigate the fight against antimicrobial resistance (AMR) in a state or nationally is mapping its infection landscape. Understanding this landscape enables building understanding of the trends in the prevalence of pathogens and hotspots and in antibiotic usage, which can be generated by integrating diverse data streams, including microbiological surveillance, clinical metadata, prescription usage patterns, and social demographics [1,2]. This facilitates the identification of resistance hotspots, guides empirical antibiotic therapy, and supports targeted interventions through antimicrobial stewardship programs and infection prevention and control (IPC) strategies. This understanding can contribute to policy interventions at two levels: one, at the level of the state, covering the entire resident population; and two, at the level of particular health facilities to enhance clinical decision-making and optimize infection management protocols.

Building such an infection landscape is particularly challenging in low- and middle-income countries (LMICs), where routine reporting of AMR infection data is minimal or even non-existent. The existing invisibility of such data needs to be first overcome through appropriate digital interventions, establishing institutional routines in health facilities to facilitate such reporting, and conducting meaningful analysis of antibiograms and enabling their dissemination within the facility to enhance visibility and action taking [3]. This paper reports on such a process from the state of Bihar in India, which has been ongoing over the last 5 years, since 2020, based on which we analyze the process of building the AMR infection landscape in Bihar’s public health system. Our analysis helps characterize this landscape and draw implications for policy and practice.

Bihar is a particularly challenging site for such an analysis for multiple reasons. One, it is a densely populated state, with a population of more than 130 million, with tremendous diversity and inequities, demanding a perspective that covers the entire state and not individual pockets [4]. Two, the state health system is currently highly compromised, and it is ranked as the second to last in the country in terms of the Human Development Index (NITI Aayog 2021) [5], reflecting high levels of poverty and illiteracy, with inequitable and suboptimal access to healthcare services. Three, the state suffers from high levels of infectious diseases, where AMR is arguably a substantial contributing factor [5]. In 2023, Bihar reported approximately 161,000 notified tuberculosis (TB) cases, making it the state with the second-highest TB burden in India. Additionally, TB-related deaths in Bihar rose by a staggering 64.1% between 2020 and 2023. The treatment success rate for newly diagnosed TB cases in Bihar was 73%, below the national average of 79.5%, indicating challenges in effective TB management (India TB Report, 2024) [6]. Leprosy is another primary concern in the state, with 11,318 active cases reported as of January 2023 and a prevalence rate of 0.77 per 10,000, higher than the national average of 0.61. Furthermore, Bihar remains endemic to multiple other infectious diseases, such as kala-azar (visceral leishmaniasis) and acute respiratory infections (ARIs), both of which significantly burden the public health infrastructure (NHSRC Health Dossier 2021) [7]. Although India has significantly reduced kala-azar cases, declining by 98.7% since 2007, Bihar still contributes more than 90% of the country’s cases. This sustained burden of infectious diseases, compounded by diagnostic constraints and the prevalent use of empirical antibiotic therapy, creates a conducive environment for the emergence and spread of AMR within the state, with an overburdened public health system ill-equipped to respond [6,7]. Left unchecked, these threats are expected to magnify in the future.

This paper aims to make the following contributions to the field of research on antimicrobial resistance (AMR), including its biomedical and digital components, with a specific focus on LMICs.

It moves beyond snapshot-based studies that focus on specific pathogens and antibiotics, adopting instead a comprehensive health system perspective that encompasses nearly all tertiary medical institutions within the state and studies them over time.

Rather than treating data as readily available for analysis, this paper highlights the foundational process of establishing data generation and reporting routines from the ground up. It emphasizes the challenges of institutionalizing these practices within a resource-constrained public health system.

This study extends beyond conventional academic analyses by also offering practical guidelines for both policy formulation and clinical practice and strengthening digital interventions.

The remainder of this paper is structured in the following way. Following this brief introduction, outlining the research aims and motivation, the next section summarizes relevant research in the domain and identifies key gaps. Section 4 describes the study methodology and describes the empirical sites and settings. Section 2 contains two parts: the first outlines the implementation of the digitally enabled processes to enable routine reporting, including the challenges experienced and the strategies employed to address them; the second presents an analysis of the characteristics of the AMR infection landscape. Section 3 discusses the key contributions of our study. Finally, we present pertinent implications for policy and practice in the conclusions (Section 5).

### 1.1. Related Research in the Context of LMICs

The prevalence of AMR is significantly higher in LMICs than in other parts of the world, yet LMICs tend to be less equipped to track and mitigate resistance developments adequately [8]. According to the WHO, most LMICs continue to lack adequate laboratory capacity, skilled personnel, standard operating procedures, digital infrastructure, and governance frameworks necessary for efficient AMR monitoring and action. These constraints lead to fragmented and occasionally unrepresentative data, which makes it more challenging to design acceptable and locally relevant guidelines for antibiotic stewardship, IPC, and empirical treatment standards [9]. Numerous studies have demonstrated that AMR data in LMICs often originate solely from tertiary-level hospitals in urban centers, which do not reflect community-level realities [10,11]. For instance, a comprehensive analysis of AMR in South Asia, published in The Lancet (Murray et al., 2022), showed that LMICs in sub-Saharan Africa and South Asia bear very high mortality burdens attributable to drug-resistant infections. The study estimated that, in 2019 alone, 1.27 million deaths were directly attributable to AMR, with 4.95 million deaths associated with drug-resistant infections [12]. Further, inadequate access to quality diagnostics and the unregulated use of antibiotics in humans, livestock, and the agricultural sectors amplify the resistance. Despite these trends, most LMICs remain underrepresented in national and global AMR data ecosystems, limiting timely and evidence-based policy and practice interventions [12].

Despite these constraints, various initiatives are underway to improve AMR data management systems in LMICs. The WHO’s Global Antimicrobial Resistance Surveillance System (GLASS) was established in 2015 to standardize AMR reporting. As of 2023, 89 countries have enrolled in GLASS, but only a subset of LMICs contribute consistently due to infrastructural and governance barriers [13]. In Southeast Asia, the “Capturing Data on Antimicrobial Resistance Patterns and Trends in Use in Regions of Asia (CAPTURA)” project has focused on extracting usable AMR data from existing hospital information systems in countries like Indonesia, Bangladesh, and Nepal. The findings from this project have informed regional planning, but the data tends to be weakly integrated with clinical workflows [14,15]. In India, national initiatives like the Indian Council of Medical Research’s AMR Surveillance Network (ICMR-AMRSN) have successfully developed pathogen–drug resistance profiles. However, underdeveloped and rural areas like Bihar are still lagging and are weakly represented in these networks. While there is a need to create longitudinal infection landscapes across healthcare levels, starting from the community, most of the data originates from hospital-based outbreaks and specific pathogen-level resistance patterns [16].

Certain LMICs have started utilizing digital technologies and decentralized monitoring techniques to fill the AMR data gap through the digitization of hospital-based dashboards, automated antibiograms, and electronic health records. However, implementation studies conducted in Kenya, Nigeria, and Nepal highlight that these digital innovations are only practical when complemented with departmental collaboration, institutional data usage practices, and workforce training initiatives. Even well-designed digital solutions may be underutilized if there is a weak culture of regular reporting, feedback, and action-taking. Furthermore, few studies have systematically documented establishing and institutionalizing these monitoring systems, particularly in public sector settings with limited resources [17]. This study seeks to address this critical gap by offering an in-depth description and evaluation of the process of deployment and gradual stabilization of digital AMR monitoring systems at five tertiary care public institutions in Bihar.

### 1.2. The Bihar AMR Context

Despite its public health significance, given its significant population and high disease burdens, Bihar remains absent from India’s AMR surveillance landscape. Previous studies carried out in the state have drawn attention to the high levels of AMR incidence, such as in *Acinetobacter* species, *Escherichia coli*, and other bacterial species, to third-generation cephalosporins and fluoroquinolones [18,19]. Another study evaluated antibiotic susceptibility (AST) patterns among hospitalized patients with advanced HIV illness in Bihar [20], highlighting widespread resistance to commonly prescribed antibiotics. According to a study conducted at Patna Medical College, multidrug-resistant *P. aeruginosa* is rising in the pediatrics, surgery, and medicine departments of the hospital [19]. However, most of these studies were conducted in distinct urban tertiary care settings, with a limited clinical focus, small sample sizes, and short follow-up times, and not contextualized within diverse demographic profiles [21]. Global systems, like Global AMR Surveillance System (GLASS), NICE (UK), and CDC (US), as well as national organizations, like the Indian Council of Medical Research (ICMR) and the National Centre for Disease Control (NCDC), highlight the necessity of standardized diagnosis, systematic resistance monitoring, and the integration of laboratory data into clinical decision-making. However, systematic shortcomings in the state reporting systems hinder the establishment of these processes.

The absence of reliable, state-level AMR data significantly undermines efforts to design evidence-based clinical and public health interventions. The flaws in the current diagnostic and research systems significantly affect public health management and planning. AMR-related delayed diagnosis and often inappropriate empirical treatment which results in more extended hospital stays, problems from sepsis, and higher mortality rates [22]. Since Bihar does not actively participate in national AMR reporting systems like NARS-Net and ICMR-AMRSN, it is more challenging to offer context-specific antimicrobial suggestions. Most recent Bihar studies lack consistency, geographical coverage, and specimen type variations. These research and structural deficiencies highlight the importance of developing a comprehensive and representative picture of the AMR landscape profile in Bihar. Building surveillance at scale is particularly important in settings with limited resources, where evidence-based prioritizing of training, drug supply, laboratory infrastructure, and stewardship operations is essential. This paper seeks to address that gap by constructing a statewide AMR infection landscape based on an analysis of AST data across multiple public tertiary hospitals. Such an analysis is relevant for developing both state- and hospital-specific treatment guidelines.

## 2. Results

### 2.1. Baseline Laboratory Capacity Assessment

The lab assessment survey identified significant quality issues arising from manual testing methods, suboptimal selection of culture media, and the absence of quality control protocols. It was noticeable that the labs conducted culture tests of less than 2% of daily outpatient visits, reflecting a high level of unmet diagnostic needs (Table 1). The baseline assessment reported the following key challenges that are summarized below.

Most labs did not use a standardized requisition form to collect essential patient details, affecting test quality and traceability.

Most of the institutions did not have a functional blood culture facility.

Culture-based microbiology testing was conducted for less than 2% of outpatient cases, indicating severe underutilization of laboratory services.

Most hospitals lacked the capacity to handle high patient loads, leading to delayed or missed diagnoses.

All the hospitals relied on paper-based or manual record-keeping.

Urine samples were often collected in glass tubes instead of sterile, leak-proof containers, increasing contamination risk and inaccurate culture results.

Improper swabbing techniques and streaking of culture plates led to inconsistent bacterial isolation, affecting AST accuracy.

Many labs had non-functional equipment and lacked the proper reagents and media required for microbial identification.

Antibiotic disc selection was inconsistent, affecting test reliability.

A shortage of trained microbiologists and technicians led to errors in diagnostic processes and poor adherence to quality control measures.

The hospitals studied lacked active Infection Control Committees (HICCs) and AMR stewardship programs, leading to unchecked empirical antibiotic use and increasing AMR risks.

### 2.2. Actions Taken to Improve Data Systems for AMR Testing and Reporting

During the research period, a number of remedial measures were implemented in order to rectify these baseline discrepancies. In all these facilities, open-source digital tools were implemented for electronic data entry and automated antibiogram generation, replacing manual record-keeping. Monthly internal reviews of AMR data and validation against physical lab registers enhanced data accuracy. The introduction of standardized requisition forms improved data quality and traceability for culture and AST reporting by ensuring that they were regularly filled out with relevant patient and clinical information. These interventions led to a progressive rise in culture testing numbers and diagnostic coverage; for instance, NMCH alone showed an increase from 2865 cultures in 2023 to 12,899 in 2024. This operational strengthening not only improved internal reporting but also enabled meaningful comparison with national ICMR-AMRSN surveillance benchmarks.

### 2.3. Geographic Distribution of AMR Diagnostic Facilities

Figure 1 illustrates the geographical distribution of essential healthcare facilities, comprising NABL-certified laboratories (blue squares), medical colleges (orange triangles), and population centers (black dots) in Bihar. These population centers include state capitals, district headquarters, and major cities and towns. While the state shows a high absolute number of diagnostic facilities, their availability is inadequate relative to its population of 130 million, with an overburdened health infrastructure and limited AMR surveillance capacity. Among the five tertiary hospitals included in this study, three are located in districts with relatively better access to diagnostic infrastructure, falling within the 1-to-2-h travel time zones. These include Patna, Muzaffarpur, and Darbhanga, which are better connected by road and transport networks and are centrally positioned within the state. In contrast, JLNMCH in Bhagalpur and adjacent eastern regions exhibit longer travel times exceeding 2 to 4 h, while certain northern and southern parts of Bihar fall within zones requiring more than 4 h to reach the nearest tertiary diagnostic facility.

Table 2 presents the total number of clinical samples tested for AST across each hospital site during the study period (2022–2024), normalized against the district-level population from the 2011 Census. This normalization allows for a population-adjusted comparison of diagnostic reach. The sample testing rate was highest at DMCH Darbhanga (308.5 per 100,000 population), followed by JLNMCH Bhagalpur (288.9) and NMCH Patna (273.4). In contrast, HBCH Muzaffarpur recorded the lowest testing rate (80.7), indicating potential under-representation or limited referral linkage at that facility.

### 2.4. Demographic Distribution of Clinical Samples Across Facilities

Table 3 presents the distribution of clinical samples tested for AST by different age groups across the five tertiary care facilities. A total of 48,456 samples were received, with the highest proportion of samples received from NMCH Patna (n = 15,787), followed by DMCH (n = 12,144) and JLNMCH (n = 8759). The 20–29-year age group accounted for the largest share of received samples, with particularly high numbers observed at NMCH (n = 6217) and DMCH (n = 4803), as shown in Table 3.

Figure 2 presents the gender-wise distribution of samples received across the five facilities. In most hospitals, more samples were received from female patients. NMCH Patna recorded the highest number of female samples (n = 12,492) compared to males (n = 3294), followed by DMCH (n = 9144 females vs. n = 3003 males) and SKMCH (n = 6318 females vs. n = 1579 males). In contrast, HBCH reported more male samples (n = 2499) than female samples (n = 1372).

### 2.5. Year-Wise Distribution of Samples Received by Different Hospital Facilities in Bihar

Figure 3 presents the number of samples received from the five hospitals during the study period. NMCH showed the highest number of samples (n = 15,764), with a rise from 2865 in 2023 to 12,899 in 2024 (350.2% increase). DMCH Darbhanga followed with 12,132 samples across the three years, showing a consistent annual growth of 18.9%. JLNMCH Bhagalpur recorded a total of 8759 samples. SKMCH Muzaffarpur contributed 7865 samples, and HBCH Muzaffarpur accounted for 3708 samples. These figures indicate a substantial increase in sample volume over time in all the facilities.

### 2.6. Sample Types Received from the Different Hospitals

Figure 4 shows the distribution of clinical samples evaluated by specimen type across the tertiary care facilities. Urine samples account for more than 90% of all hospital specimens. The second most common sample type among these facilities is related to superficial infections. HBCH displayed a comparatively greater percentage of blood samples, 2436 (63.4%), followed by superficial infections, 414 (10.8%).

### 2.7. Number of Samples Received from Different Departments

The distribution of clinical samples throughout the various hospital departments at the different tertiary care facilities is depicted in Figure 5. The Obstetrics and Gynecology (OBG) department provided the most significant percentage of samples at almost all facilities, accounting for more than half of the total, including 61.1% at NMCH Patna and 52.3% at JLNMCH Bhagalpur. For example, the Pediatrics department contributed 9.7% at DMCH and 13.4% at SKMCH. In contrast, the Medicine department comprised 18.0% at HBCH&RC and 28.8% at SKMCH. The sample contributions from departments including Urology, ENT, Dermatology, Neurosurgery, Ophthalmology, and the Burns unit was consistently low, frequently less than 1%.

### 2.8. Distribution of Bacterial Isolates from Clinical Samples

The distribution of bacterial species isolated from urine and superficial infection and blood samples among the different facilities is shown in Figure 6. The most prevalent bacterial species isolated from urine samples was *E. coli*, whereas the most prevalent bacterial species found in samples of superficial infections was *S. aureus*.

### 2.9. Facility-Wise Variation in Susceptibility Patterns from Clinical Isolates

Figure 7 presents a comparative analysis of antibiotic sensitivity patterns in *E. coli* isolates from urine samples across the five facilities. Sensitivity to Nitrofurantoin was highest at NMCH (86.51%) and HBCH (85.71%) but markedly lower at JLNMCH (44.72%). Amikacin and Gentamicin demonstrated high sensitivity at SKMCH (95.07% and 99.62%, respectively), indicating their continued efficacy. Meropenem and Imipenem exhibited high sensitivity at NMCH (80.00% and 73.29%, respectively), but sensitivity dropped significantly at HBCH. Fluoroquinolones, including Ciprofloxacin and Levofloxacin, showed low sensitivity across most sites, particularly at SKMCH and HBCH. Beta-lactam antibiotics, such as Cefotaxime, Ceftriaxone, and Ampicillin, showed poor sensitivity, reflecting widespread resistance. In contrast, Fosfomycin displayed high sensitivity at DMCH (84.42%) and HBCH (97.14%), though it was not tested elsewhere.

Figure 8 presents inter-facility variations in antibiotic sensitivity of *Staphylococcus aureus* isolates obtained from superficial infection samples. Gentamicin and Amikacin exhibited consistently high sensitivity across most sites, with values exceeding 70% at all reporting hospitals, and the highest sensitivity was observed for Amikacin at JLNMCH (90.00%). Azithromycin showed high sensitivity at SKMCH (86.28%) and NMCH (69.44%) but was notably low at DMCH (25.76%). Levofloxacin sensitivity ranged widely, from 12.06% at DMCH to 68.42% at JLNMCH. Linezolid sensitivity showed variability, with moderate activity at NMCH (58.33%) and HBCH (69.23%) but lower at SKMCH (42.47%). Ciprofloxacin and Erythromycin demonstrated poor sensitivity across all centers, with the lowest values reported at HBCH. JLNMCH showed relatively better sensitivity to Clindamycin (78.85%) and Norfloxacin (66.10%). Vancomycin sensitivity was moderate at HBCH (59.38%) but lower at JLNMCH (45.45%).

In *Klebsiella* spp., gentamicin and amikacin show the highest efficacy, particularly at SKMCH (98.86% and 95.09%) and JLNMCH (86.11% and 87.34%). Sensitivity to commonly used antibiotics like ciprofloxacin, cefotaxime, and ampicillin remains critically low across all sites (Figure 9).

## 3. Discussion

To characterize the AMR infection landscape in the state, we conducted a comparative analysis of AMR patterns across facilities in Bihar and compared them with some national trends in India and other LMICs. Based on this mapping, we develop some important implications for policy and clinical practice.

### 3.1. AMR Trends Within and Across Facilities

The present study provides a comprehensive depiction of AMR data patterns within the public health systems of an underprivileged Indian state, Bihar. The amalgamation of three years of microbiological and clinical data from five prominent tertiary care hospitals offers a unique, facility-specific analysis of resistance trends among pathogens. This study finds substantial resistance trends, particularly among WHO priority pathogens, including *E. coli*, *Klebsiella* spp., and *S. aureus*, while emphasizing the systemic inequities in diagnostic capacity and stewardship readiness that influence the AMR landscape. It is unique for several hospitals in a single LMIC state to undergo such assessments, and our results fill important data gaps in strengthening provincial and national monitoring systems in India.

Our study reports relatively high sensitivity of *E. coli* to Nitrofurantoin at NMCH (86.5%) while showing a dramatic fall in potency at JLNMCH (44.7%) and SKMCH (61.39%). Our study revealed extremely high resistance in *Klebsiella* spp. to third-generation Cephalosporins, with Cefotaxime and Ceftriaxone sensitivities dropping as low as 0.56% and 1.91%, respectively, at SKMCH, alongside Ciprofloxacin sensitivity of just 8.2%. Carbapenems, including Imipenem, Meropenem, and Ertapenem, are considered last-resort antibiotics for treating multidrug-resistant (MDR) Gram-negative infections. In our study, their efficacy showed considerable inter-facility variation, with Imipenem sensitivity in *E. coli* ranging from 73.29% at NMCH to lower values at HBCH (50%), while Meropenem efficacy declined to 57.1% at HBCH as compared to NMCH (80%). There were similar patterns of rising resistance in commonly prescribed antibiotic categories. Amikacin and Gentamicin showed comparatively high sensitivity across facilities, particularly at SKMCH (95.1% and 99.6%, respectively). However, Cefoxitin sensitivity, a proxy for Methicillin-resistant *S. aureus* (MRSA) detection, was significantly higher. The sensitivity of cefoxitin was very low in the two hospital settings, i.e., 25.3% at DMCH and 34.0% at HBMCH, corresponding to MRSA prevalence rates of about 75% and 66%, respectively. These results emphasize the urgent need for effective stewardship protocols and facility-specific treatment recommendations to better manage an array of resistance trends and ensure effective antibiotic usage across all Bihar healthcare settings.

### 3.2. Comparison with National and LMICs AMR Trends

Comparative examination of data on AST from five Bihar-based tertiary care centers with national-level results from the ICMR-AMRSN surveillance (2023) [23] depicts substantial regional heterogeneity in resistance trends. There are points of overlap: for instance, nationally, ICMR-AMRSN reported Nitrofurantoin susceptibility at 85.8%, suggesting that while some Bihar facilities align with national trends, others reflect higher levels of localized resistance [23]. The multifactorial nature of AMR is highlighted by the possibility that this variance may result from laboratory procedures, population demographics, or the presence of resistant clones. For *S. aureus* isolates from pus samples, the national data shows excellent efficacy of Linezolid (97.7%), Clindamycin (77.1%), and Vancomycin (100%). The reduced Linezolid susceptibility at NMCH (58.3%) and SKMCH (42.5%), however, may be a result of rising resistance, perhaps spurred on by the overuse of antibiotics. However, it could also reflect variability in disc potency, storage conditions, or testing methodology. Clindamycin maintained its efficacy at JLNMCH (78.9%), consistent with national trends (77.1%). The national burden of MRSA, as reported by ICMR-AMRSN, was 47.8%. The overall trend of MRSA in two hospitals in Bihar, however, indicates a high MRSA burden [23]. These findings highlight the need for region-specific AMR strategies alongside national efforts. Our results highlight the vulnerability of widely used antibiotic classes. Common drugs like cephalosporins and fluoroquinolones showed very low sensitivity across facilities, indicating declining effectiveness and the need for data-driven prescribing.

In Bihar facilities, the Amikacin and Gentamicin sensitivity was higher than the ICMR-AMRSN 2023 report, where Amikacin sensitivity for *E. coli* was 71.1%, and for *Klebsiella* spp., it was 44.4%. This suggests that, while medications are still effective treatment alternatives in many Indian hospital settings [23], this may not be the case in the state. These findings closely mirror those of Gandra et al. (2017), who reported widespread resistance to third-generation Cephalosporins and Fluoroquinolones in *E. coli* and *Klebsiella* across Indian tertiary hospitals [24]. Similarly, the majority of LMICs include cephalosporin-resistant *Klebsiella* (>70%), according to WHO GLASS (2022) surveillance data [25,26,27,28]. In line with studies conducted in Bangladesh and Nepal, Ciprofloxacin resistance was highest across Bihar tertiary care facilities, indicating broad regional Fluoroquinolone resistance based on significant empirical usage [29,30]. Our study reported high levels of Fluoroquinolone and beta-lactam resistance, similar to a study in Tamil Nadu, which found that urine isolates had a resistance rate of >75% to Cefotaxime and Ceftriaxone. This was similar to what we found at SKMCH and HBCH [31]. According to Islam et al. (2022), Fluoroquinolone resistance in Bangladeshi outpatient settings was above 80%. This may be due to regional antibiotic overuse and restricted access to diagnostics [32]. Our findings on high resistance in *Klebsiella* and *S. aureus* are consistent with Shen et al. (2021), who reported similar resistance patterns in rural China [33]. Due to the impact of non-optimal use of antibiotics, Azithromycin was less effective in *S. aureus* at DMCH (25.8%) but substantially more effective at SKMCH (86.3%). Regional studies from Nepal and SEARO countries show similar trends, warning of increasing resistance even among *S. aureus* strains that are resistant to Linezolid and of variations in antibiotic effectiveness for UTIs in hospitals in South-East Asian LMICs due to local prescribing patterns [34,35]. According to ICMR-AMRSN 2023, *E. coli* susceptibility to Imipenem was 72.0%, and to Meropenem, it was 77.2% nationally [23]. This suggests that Bihar facilities are starting to follow national patterns, as demonstrated by NMCH, but that Carbapenem resistance continues to develop at HBCH. The differences at the HBCH site could also be due to the selection of strains and a smaller number of isolates in general. The decreasing effectiveness of Carbapenems highlights the limited window of treatment for infections acquired in hospitals, particularly in intensive care units (ICUs). These findings underscore the need for strict stewardship, regular monitoring, and effective infection control measures to maintain the therapeutic efficacy of Carbapenems. Overall, these discrepancies in AMR trends can be the result of regional consumption of antibiotics trends, irregularities in the supply chain, or monitoring gaps in stewardship. This inter-facility diversity in resistance trends brings significant concerns about local antibiotic prescription practices, drug procurement heterogeneity, and stewardship protocols.

### 3.3. Beyond Pathogens: Social and Spatial Drivers of AMR

This study provides a comprehensive assessment of resistance trends in different pathogens and institutions, with a focus on WHO priority pathogens, like *E. coli*, *Klebsiella* spp., and *S. aureus*. However, the resistance trends were not limited to these pathogens. In institutions such as HBCH, isolates such as *P. aeruginosa* and *Enterococcus* spp. were also identified, indicating an increased AMR load among Gram-negative and Gram-positive bacteria. Although their number was limited, these organisms possessed explosive resistance profiles, raising the need for expanded surveillance beyond core pathogens. This analysis is not only inter-facility but also strongly related to demographic and departmental factors. Most of the samples were from young adults (20–29 years), were from Gynecology departments (e.g., >60% at NMCH), and are likely to represent gendered healthcare seeking and institutional sampling biases. The demographic bias we observed affects both the detection of pathogens and resistance patterns. For example, high empirical use of antibiotics in Surgical and Maternity wards may be due to the high resistance to commonly used drugs amongst these patient groups. This heterogeneity highlights the need to contextualize antibiograms along with demographic and clinical metadata while developing facility-specific treatment policies. In addition, spatial disparities in diagnosis accessibility are likely to underlie geographical variation in observed AMR trends. Our study highlighted hospital locations and travel access as crucial in identifying infection types and resistance hotspots. Mapping of these trends against regional health infrastructure and disease burden can further explain how geographic disparities generate AMR variation and contribute to building informed and locally relevant stewardship interventions.

### 3.4. Implications for Policy and Clinical Practice

This study highlights how integrating digitally generated AMR data into routine hospital workflows in Bihar is both feasible and impactful, despite significant infrastructural constraints. Based on our findings and through comparative insights drawn from Himachal Pradesh (HP), where similar interventions have been piloted, we identify distinct policy and clinical practice implications to guide the future of AMR surveillance and stewardship in LMIC public health systems more broadly. Based on our findings, we propose a focused and prioritized policy framework to strengthen AMR containment efforts in Bihar and similar LMIC settings.

**1.** 
**Real-time Digital Surveillance**


Establish a state-level AMR dashboard that enables microbiologists to routinely upload, access, and review facility-level antibiograms. This will enhance data visibility, promote real-time validation, and support timely interventions at both the institutional and state levels.

**2.** 
**Data-Informed Procurement and EML Alignment**


Ensure that the procurement of antibiotics and updates to the EML drugs are guided by local susceptibility trends. This alignment will reduce the stockpiling of ineffective drugs and improve clinical outcomes by making effective antibiotics readily available.

**3.** 
**Strengthening Institutional Stewardship Mechanisms**


Mandate interdepartmental AMR audits, feedback loops between microbiologists and clinicians, and routine review of empirical antibiotic use based on local resistance data. These practices will institutionalize accountability and foster rational prescribing.

**4.** 
**Improving Access to Diagnostics**


Address geographic and infrastructural barriers by strengthening laboratory capacity in underserved areas. GIS-based planning should be used to guide investments and ensure equitable access to microbiology services across the state.

## 4. Material and Methods

### 4.1. Study Sites and Catchment Area Demographics

This research was conducted in five government tertiary care hospitals in Bihar: Nalanda Medical College and Hospital (NMCH), Patna; Sri Krishna Medical College and Hospital (SKMCH), Muzaffarpur; Jawaharlal Nehru Medical College and Hospital (JLNMCH), Bhagalpur; Darbhanga Medical College and Hospital (DMCH), Darbhanga; and Homi Bhabha Cancer Hospital (HBCH), Muzaffarpur (Figure 10). These facilities each have extensive catchment areas, serving urban and rural populations with diverse healthcare needs. Each hospital has an operational microbiology laboratory tasked with processing clinical specimens and routine diagnostic testing, including culture and antimicrobial susceptibility testing (AST).

The hospitals studied highlight diverse socio-demographic and epidemiological characteristics of their catchment areas, including population size, rural composition, health infrastructure, and disease burden (Table 4). Darbhanga and Muzaffarpur are primarily rural and more dependent on public healthcare. The health facilities of Bhagalpur are extremely inadequate, particularly related to readily accessible diagnostics.

### 4.2. Study Timeline

This study took place over 3 years (1 January 2022 to 31 December 2024), and data were continuously collected from the microbiology labs across all five hospitals during this period.

### 4.3. Study Team

This study was led by a multidisciplinary team comprising microbiologists, clinicians, data scientists, and public health researchers from the participating hospitals and collaborating institutions. Designated microbiology teams at each site were responsible for sample collection, culture, and antimicrobial susceptibility testing following standard laboratory protocols. Clinical teams supported data entry and verification. The microbiologists were engaged with the development of the evolving AMR infection profile within their respective hospitals.

### 4.4. Spatial Analysis of Diagnostic Access

Further, a spatial analysis was conducted to assess physical access to microbiology diagnostic services across Bihar. Public tertiary care institutions with microbiology laboratories and National Accreditation Board for Testing and Calibration Laboratories (NABL) accredited labs were geolocated and mapped against population density and transport infrastructure. Geographic information system (GIS) tools were used to estimate travel time and distance from peripheral districts to the nearest functional diagnostic facility. A map showing the geographic locations and catchment areas of the five selected tertiary care hospitals in Bihar is provided in the results Section 2.

Spatial analyses were conducted using R software version 4.2.2. Geocoding was performed via the Google Geocoding API, with validation through OpenStreetMap reverse geocoding. Driving distance and time were estimated using the Google Distance Matrix API. Spatial interpolation methods, including Voronoi polygons, k-nearest neighbor (k-NN), and inverse distance weighting (IDW), were implemented under the projected coordinate reference system EPSG:7755 (India 2000^TM^) to ensure accuracy.

### 4.5. Baseline Laboratory Assessment

At the initiation of this study, a baseline assessment of microbiology laboratories was conducted across nine government medical colleges in Bihar. A standardized data collection tool evaluated laboratory infrastructure, testing capacity, human resources, and workflow processes. Field visits involved on-site observations, semi-structured interviews with laboratory personnel, and a review of microbiology registers. The assessment covered core areas such as equipment availability, sample handling protocols, AST procedures, use of requisition forms, quality control practices, biosafety measures, and the extent of digitization in result reporting. Key findings from this assessment are summarized in the results Section 2.

### 4.6. Clinical Data

The microbiology laboratory in each hospital received about 15–20 samples a day for testing, including urine, pus exudate, and other samples from patients attending the outpatient department (OPD) and the inpatient department (IPD). Blood culture samples were noticeably minimal in the current testing profiles across all hospitals. The samples were sent in for culture tests, and reports were generated for the patient and the doctor requesting further treatment guidance.

### 4.7. Clinical Sample Processing and Bacterial Identification

All clinical samples were processed in the microbiology laboratory using conventional culture techniques. All the culture media and AST discs were taken from HiMedia Laboratories Pvt. Ltd., Mumbai, India. The urine samples were inoculated on cysteine lactose electrolyte deficient (CLED) agar medium. Inoculated agar plates were incubated initially aerobically at 37 °C for 24 h and, finally, for 48 h, and the plates were examined for pure growth. A growth of ≥10^5^ colony-forming units/mL was considered significant. Cultures with more than two types of colonies were considered contaminants, and such samples were discarded. Pus samples were processed for Gram staining and culturing. The samples were aseptically inoculated on blood agar (with 5% sheep blood) and MacConkey agar plates and incubated aerobically at 35–37 °C for 24–48 h. For the conventional blood culture method, blood culture for bacterial infections was carried out in two bottles containing 50 mL each of tryptone soy broth and bile broth. After removing the Kraft paper, the culture bottles were inoculated. These were incubated at 37 °C and examined daily for 7 days for evidence of growth, which was indicated by turbidity, hemolysis, gas production, discrete colonies, or a combination of these. The broth from the positive bottle was subcultured onto 5% sheep blood agar, and MacConkey plates were incubated similarly for the pus samples. Pathogens were identified based on colony morphology, Gram staining, and standard biochemical tests. The primary focus was on isolating WHO-designated critical priority pathogens for further analysis [36].

### 4.8. Antibiotic Susceptibility Testing (AST)

AST was performed with the disc diffusion method for the following antimicrobials: doxycycline, azithromycin, ampicillin, ciprofloxacin, amikacin, imipenem, levofloxacin, gentamicin, cefepime, piperacillin–tazobactam, ertapenem, meropenem, cotrimoxazole, cefoxitin, ceftriaxone, etc., according to Clinical & Laboratory Standards Institute (CLSI) guidelines [37,38]. The antibiotic discs were selected based on CLSI guidelines and the specific organism being tested to ensure appropriate pathogen–drug matching. The national data from the ICMR Antimicrobial Resistance Surveillance Network (AMRSN) were used as a benchmark to compare susceptibility patterns observed in five tertiary care hospitals in Bihar [23].

### 4.9. Digital Integration Process and Observational Insights

A digital integration workflow was implemented across each of the participating hospitals to support standardized reporting and streamline data flow around the microbiology testing process, including sample recording, documenting of test results, and analysis of infection patterns. The digital system was based on the baseline assessment conducted, which helped identify the testing and reporting infrastructure, existing challenges, and opportunities for improvement [39]. Each microbiology laboratory initially maintained paper-based records, which were then digitized using a customized design on an open-source software platform to capture patient demographics, sample type, pathogen isolated, and antibiotic susceptibility test results. Data entry was conducted by laboratory personnel who were trained by the research team, and data quality was periodically reviewed to ensure accuracy and completeness. A prototype electronic AMR dashboard was created, allowing for real-time data visualization and automated antibiogram generation. The system was developed using open-source tools, and monthly/quarterly facility-level data reviews were allowed by nodal microbiologists. Validation of the digital data involved cross-checking a subset of entries against physical laboratory registers to assess consistency and reduce transcription errors. In addition to system implementation, the research team conducted field observations at all five sites to understand ground-level practices, variations in digital readiness, challenges in specimen tracking, irregular use of request forms, and discrepancies in data reporting formats. Continuous support and capacity-building exercises were carried out to ensure the evolution of the digital application in responding to the emerging needs of microbiologists and clinicians.

### 4.10. Statistical Analysis

All data were curated, analyzed, and visualized using Microsoft Excel. Descriptive statistics, including absolute frequencies and percentages, were employed to summarize patient demographics, clinical sample characteristics, bacterial isolates, and AST patterns across the five tertiary care hospitals. Data analysis was conducted at multiple levels: within-facility assessments examined age, gender, and departmental distribution of samples and resistance patterns; inter-facility comparisons were used to evaluate variations in diagnostic practices and susceptibility trends across the five hospitals; and a comparative analysis was performed against national-level data from the ICMR-AMRSN surveillance network to contextualize the AMR landscape in Bihar within broader national infection trends.

## 5. Conclusions

Our study is the first of its kind from India to create such a broad, facility-associated AMR picture over time at a state level. This study highlights the need for building localized antimicrobial stewardship to address diagnostic disparities and encouraging department-level involvement. These results provide a roadmap for integrating diagnostic stewardship into standard clinical practice in low-resourced environments. Adapting treatments to institutional circumstances rather than depending exclusively on national statistics is crucial, as evidenced by the reported resistance diversity between institutions. Both targeted digital solutions and institutional adjustments are needed to close the gap between laboratory findings and prescription practices. Promoting more sensible antibiotic usage is possible by enabling microbiology laboratories to produce meaningful data and ensuring that it is incorporated into clinical decision-making. These revelations pave the way for long-term AMR control plans in hospital environments with limited resources.

## Figures and Tables

**Figure 1 antibiotics-14-00684-f001:**
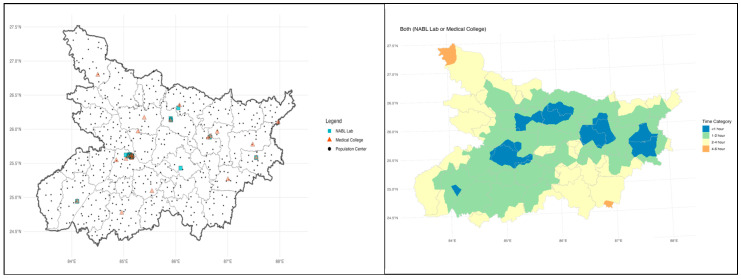
Spatial distribution of AMR diagnostic facilities and access times in Bihar.

**Figure 2 antibiotics-14-00684-f002:**
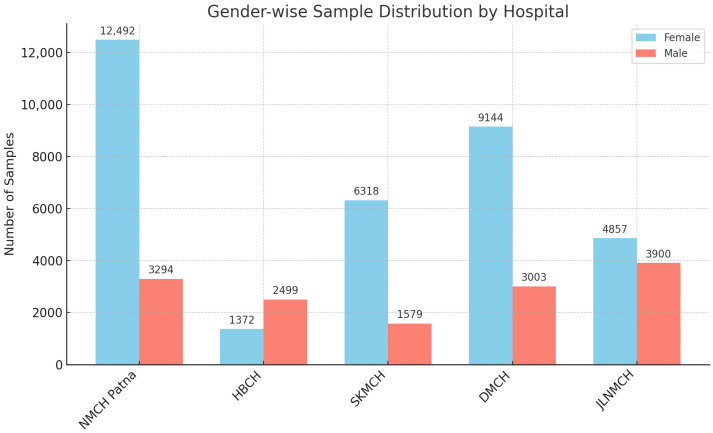
Gender-wise distribution of samples received in different hospitals in Bihar.

**Figure 3 antibiotics-14-00684-f003:**
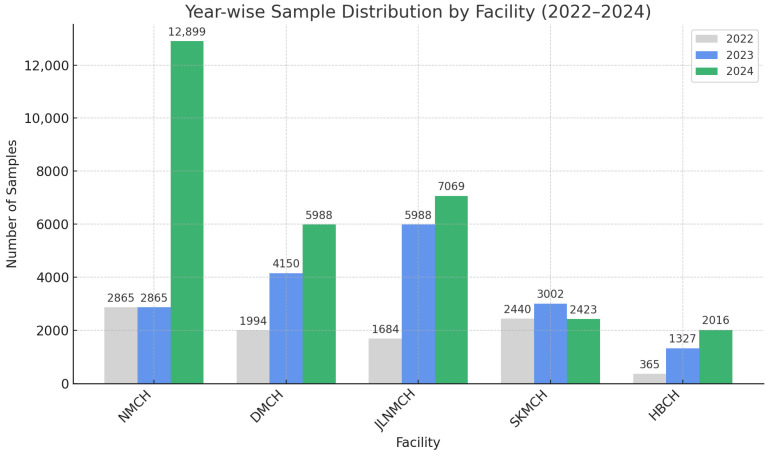
Number of samples received from different facilities.

**Figure 4 antibiotics-14-00684-f004:**
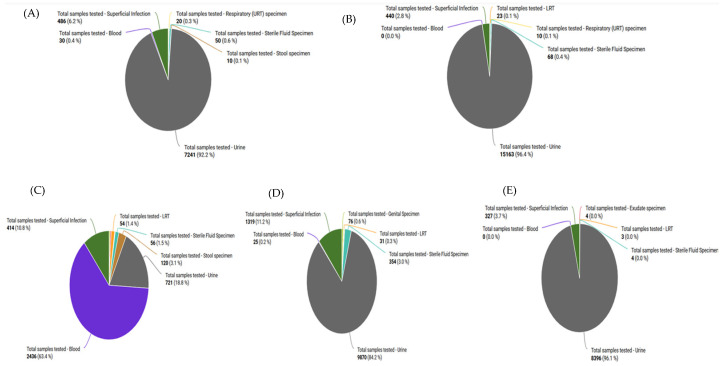
Sample types received from different hospitals in Bihar: (**A**) SKMCH, (**B**) NMCH, (**C**) HBCH, (**D**) DMCH, (**E**) JLNMCH.

**Figure 5 antibiotics-14-00684-f005:**
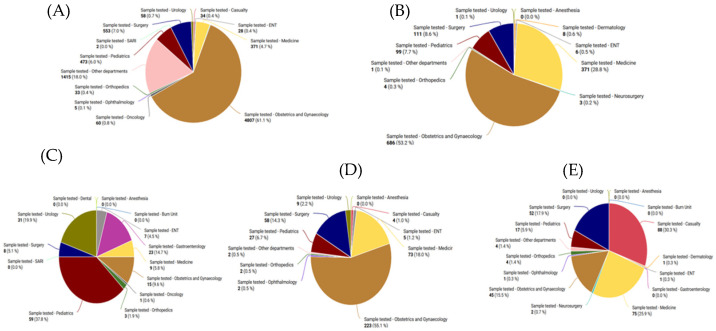
Number of samples received from different hospitals. (**A**) SKMCH, (**B**) NMCH, (**C**) HBCH, (**D**) DMCH, (**E**) JLNMCH.

**Figure 6 antibiotics-14-00684-f006:**
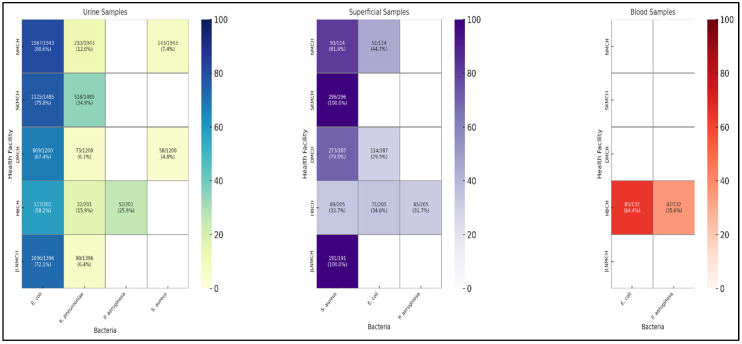
Distribution of bacterial isolates from clinical samples.

**Figure 7 antibiotics-14-00684-f007:**
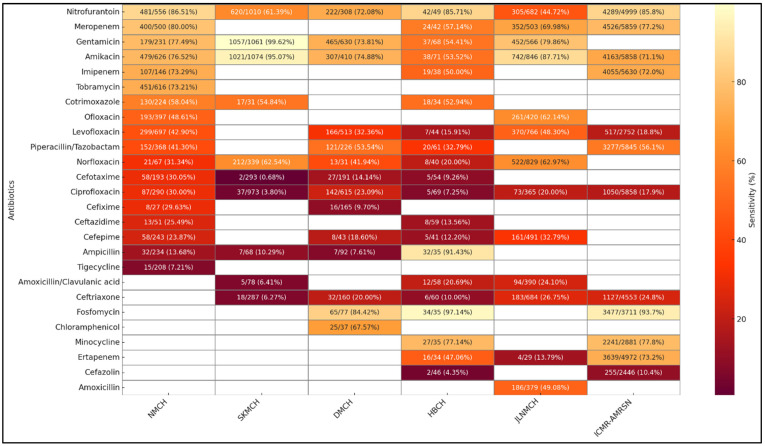
Comparative analysis of *E. coli* antibiotic susceptibility from urine samples: hospital data vs. ICMR-AMRSN surveillance.

**Figure 8 antibiotics-14-00684-f008:**
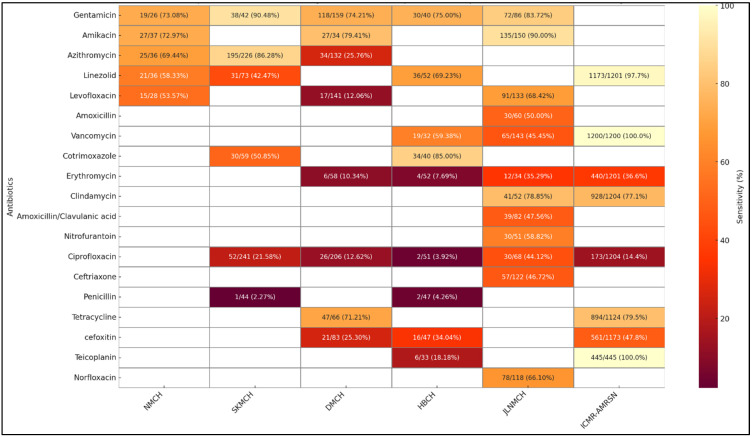
Comparative analysis of *S. aureus* antibiotic susceptibility from superficial samples: hospital data vs. ICMR-AMRSN surveillance.

**Figure 9 antibiotics-14-00684-f009:**
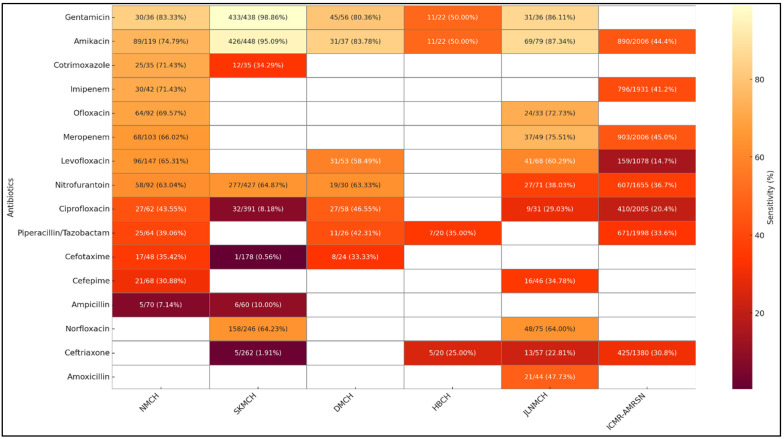
Comparative analysis of *Klebsiella* spp. antibiotic susceptibility from urine samples: hospital data vs. ICMR-AMRSN surveillance.

**Figure 10 antibiotics-14-00684-f010:**
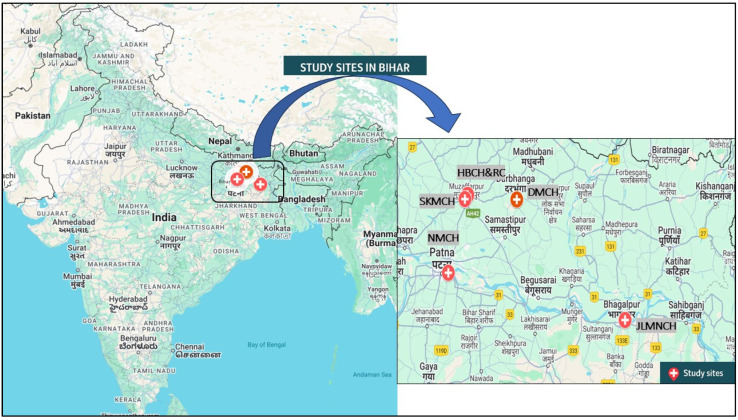
Geographic distribution of the five tertiary care hospitals included in the state.

**Table 1 antibiotics-14-00684-t001:** Status of diagnostic testing capacities in the hospitals.

Hospital Sno	Daily Patient Load	Samples Tested/Day	Testing Rate * (%)	Reporting Digital/Manual	Use of Data in the Hospital
DMCH, Darbhanga	1800	20–25	1.25	Manual	No
SKMCH, Muzaffarpur	1800–2000	15–20	0.9	Manual	No
PMCH, Patna	1500–3000	60–80	3.1	Manual	No
ANMCH, Gaya	1000	17–18	1.75	Manual	Yes
JLNMC, Bhagalpur	1500	30–35	2.17	Manual	No
JNKTMCH	650	No data	--	Manual	No
IGIMS, Patna	3000	100–150	4.17	Digital	Yes
VIMS, Pawapuri	2000	15–20	1.0	Manual	No
GMC, Bettiah	600	No data	--	Manual	No

* Testing Rate (%) represents the percentage of patients visiting each hospital who actually received microbiological testing. It is calculated by dividing the number of samples tested daily by the daily patient load and multiplying by 100. Abbreviations: DMCH: Darbhanga Medical College and Hospital, SKMCH: Sri Krishna Medical College and Hospital, PMCH: Patna Medical college and Hospital, ANMCH: Anugrah Narayan Magadh Medical College and Hospital, JLNMC: Jawaharlal Nehru Medical College, JNKTMCH: Jannayak Karpoori Thakur Medical College and Hospital, IGIMS: Indira Gandhi Institute of Medical Science, VIMS: Vardhman Institute of Medical Sciences, GMC: Government Medical College.

**Table 2 antibiotics-14-00684-t002:** Clinical samples tested relative to the catchment population (2022–2024).

Hospital	District	Samples Received	Population (2011)	Samples per 100,000
DMCH	Darbhanga	12,144	3,937,385	308.5
HBCH	Muzaffarpur	3871	4,801,062	80.7
SKMCH	Muzaffarpur	7895	4,801,062	164.4
JLNMCH	Bhagalpur	8759	3,032,226	288.9
NMCH	Patna	15,787	5,772,804	273.4

**Table 3 antibiotics-14-00684-t003:** Age-wise distribution of samples received in different hospitals.

Age Group (Years)	DMCH	HBCH	SKMCH	JLNMCH	NMCH
0–9	844	807	584	888	1306
10–19	1909	509	1018	1222	3335
20–29	4803	215	2897	2125	6217
30–39	1882	438	1789	1541	2350
40–59	1945	1123	1256	1996	2002
60–79	724	748	324	895	544
>80	37	31	27	92	33
Total	12,144	3871	7895	8759	15,787

**Table 4 antibiotics-14-00684-t004:** Socio-demographic and epidemiological indicators * of hospital catchment areas.

District	Population (2011 Census)	% Rural Population	No. of PHCs	No. of CHCs	Major Disease Burden (%)
Patna	5,772,804	57.6%	84	14	Diarrhea (7%), TB (3.6%), LRI (10.4%)
Muzaffarpur	4,801,062	89.6%	102	17	Diarrhea (8%), TB (3.9%), LRI (9.8%)
Bhagalpur	3,032,226	79.5%	65	11	Diarrhea (6.8%), TB (4.1%), LRI (8.7%)
Darbhanga	3,937,385	85.5%	90	15	Diarrhea (7.9%), TB (4.5%), LRI (9.2%)

* Sources: Census 2011 and National Health Systems Resource Centre (NHSRC) Health Dossier 2021.

## Data Availability

The datasets used and/or analyzed during the current study are available from the corresponding author upon reasonable request.

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
