# Peer review of "Mapping the AMR Infection Landscape in Bihar: Implications for Strengthening Policy and Clinical Practice"

_antibiotics, 2025, doi:10.3390/antibiotics14070684_

Round 1
Reviewer 1 Report
Comments and Suggestions for Authors
Title: “Mapping the AMR infection landscape in Bihar: Implications for strengthening policy and clinical practice”
Date: 19th June 2025
Summary
This manuscript presents a vital and timely study that evaluates antimicrobial resistance (AMR) surveillance and diagnostic capacity in Bihar, India, utilising both traditional and digital methods. The study includes qualitative assessments of laboratory capacities, digital system deployment, and policy implications, as well as empirical data from several hospitals. It emphasises the need for standardised data collection, digital integration, and stewardship programmes in combating AMR, particularly in low-resource settings. I would recommend a minor revision to address several concerns and questions.
This manuscript provides a valuable contribution to understanding and enhancing AMR surveillance and antimicrobial stewardship in resource-limited settings. By addressing the issues of clarity, improving the structure of the methodology, and explicitly acknowledging limitations, the paper can be further strengthened. I recommend minor to moderate revisions for clarity, organisation, and more detailed discussion.
Overall
- Please provide the names of the manufacturers, cities, and countries of any chemicals/media/kits/software you used in this study. Please check and correct it throughout the manuscript.
- “Geographic Information System (GIS) tools were used to estimate travel time and distance from peripheral districts to the nearest functional diagnostic facility”, page 229. Can you provide the version of the tool?
- Can you use another reference for CLSI, not these two? They are not suitable for this method. “AST was performed with the disc diffusion method for the following antimicrobials: doxycycline, azithromycin, ampicillin, ciprofloxacin, amikacin, imipenem, levofloxacin, Gentamicin, cefepime, piperacillin-tazobactam, ertapenem, meropenem, cotrimoxazole, cefoxitin, ceftriaxone, etc., according to Clinical & Laboratory Standards Institute (CLSI) 273
- guidelines [24,25]”
- Several sentences are lengthy and complex, which hampers understanding. For example, sentences like: "Building surveillance at scale is particularly important in settings with limited resources, where evidence-based prioritising of training, drug supply, laboratory infrastructure, and stewardship operations is essential, a gap which this paper seeks to address by building a statewide AMR infection landscape based on an analysis of AST across multiple public tertiary hospitals" could be split for clarity.
- Recommendation: Rewrite long, compound sentences into shorter, clearer statements. Follow a logical flow to enhance readability.
- 6. Policy and Practical Implications: The manuscript offers numerous policy recommendations, some of which are repetitive (e.g., the importance of digital dashboards). Consider consolidating similar points and synthesising the key policy actions into a concise framework, emphasising prioritised strategies for Bihar.
Comments on the Quality of English Language
The English was okay, just a few typos were errors.
Author Response
Comments:
- Please provide the names of the manufacturers, cities, and countries of any chemicals/media/kits/software you used in this study. Please check and correct it throughout the manuscript.
Ans. Thank you for the suggestion. The names of the manufacturers and software tools used in the study have now been added at appropriate places in the Materials and Methods section. Changes have been made on lines 234-239 and 259-260.
- “Geographic Information System (GIS) tools were used to estimate travel time and distance from peripheral districts to the nearest functional diagnostic facility”, page 229. Can you provide the version of the tool?
Ans. Thank you for your suggestion. The version of the GIS tools used in this study has been specified in the manuscript. Changes have been made on lines 234-239.
- Can you use another reference for CLSI, not these two? They are not suitable for this method. “AST was performed with the disc diffusion method for the following antimicrobials: doxycycline, azithromycin, ampicillin, ciprofloxacin, amikacin, imipenem, levofloxacin, Gentamicin, cefepime, piperacillin-tazobactam, ertapenem, meropenem, cotrimoxazole, cefoxitin, ceftriaxone, etc., according to Clinical & Laboratory Standards Institute (CLSI) 273 guidelines [24,25]”.
Ans. Thank you for highlighting this issue. The previous CLSI references [24,25] have been replaced with more appropriate references in the revised manuscript.
- Several sentences are lengthy and complex, which hampers understanding. For example, sentences like: "Building surveillance at scale is particularly important in settings with limited resources, where evidence-based prioritising of training, drug supply, laboratory infrastructure, and stewardship operations is essential, a gap which this paper seeks to address by building a statewide AMR infection landscape based on an analysis of AST across multiple public tertiary hospitals" could be split for clarity.
Ans. Thank you for the suggestion. We agree that sentence complexity may affect clarity. The manuscript has been revised to split lengthy sentences into shorter, clearer ones where appropriate. Please refer to lines 184-190, which are highlighted.
- Policy and Practical Implications: The manuscript offers numerous policy recommendations, some of which are repetitive (e.g., the importance of digital dashboards). Consider consolidating similar points and synthesising the key policy actions into a concise framework, emphasising prioritised strategies for Bihar.
Ans. Thank you for this valuable suggestion. We have revised the Policy and Clinical Implications section to remove repetition, especially around digital dashboards. Related points have been consolidated, and a concise framework highlighting key, prioritised strategies for Bihar has been presented for clarity and impact. Please refer to lines 597-623.
Reviewer 2 Report
Comments and Suggestions for Authors
Title: Mapping the AMR infection landscape in Bihar: Implications for strengthening policy and clinical practice
The study offers an overview of antimicrobial resistance (AMR) trends at the state level. It illustrates the efficacy of localized antibiograms in directing empirical therapy within resource-constrained environments. This research introduces a scalable framework for embedding AMR surveillance into the routine workflows of health systems in low- and middle-income countries. This manuscript contributes significant information to the field of AMR.
I have minor comments
- Line 255: ‘ml’ should be ‘mL’
- Line 270: Authors should mention how the panel of antibiotics was decided for the AST?
- Line 273 and 331: These two lines are contradictory. Authors should indicate whether they have repeated the AST according to the Clinical and Laboratory Standards Institute (CLSI) guidelines. Also, mention the edition of the CSLI guidelines followed.
- Figure 4: Revise Figure 4 for clarity. What is ‘HBCH’?
- Figures 5 and 6: The resolution of the figures is low, making them hard to read.
- Section 3.8: Year-wise distribution of the bacterial isolates should also be provided.
- Section 3.9: In addition to comparative analysis with the ICMR-AMRSN Surveillance, the AST data with respect to the bacterial isolates collected in the study should be provided. This will provide an idea of the AMR status concerning each study facility, as well as whether there is a decline or increase in AMR during the study period.
Author Response
Q1. Line 255: ‘ml’ should be ‘mL’
Ans. Thank you for pointing this out. The unit has been corrected from “ml” to “mL” in line 255, following standard scientific notation. Please refer to lines 263 and 269.
Q2. Line 270: Authors should mention how the panel of antibiotics was decided for the AST?
Ans. Thank you for the suggestion. A clarifying sentence has been added in the revised manuscript stating that the antibiotic discs used for AST were selected based on CLSI guidelines and tailored to the organism being tested, ensuring appropriate drug-pathogen matching for each isolate. Please refer to lines 282-284.
Q3. Line 273 and 331: These two lines are contradictory. Authors should indicate whether they have repeated the AST according to the Clinical and Laboratory Standards Institute (CLSI) guidelines. Also, mention the edition of the CSLI guidelines followed.
Ans. Thank you for pointing this out. Text has been modified in the revised manuscript for better clarity. Please refer to line number 341.
Q4. Figure 4: Revise Figure 4 for clarity. What is ‘HBCH’?
Ans: Figures 3 and 4 has been revised for better clarity in the revised manuscript.
Q5. Figures 5 and 6: The resolution of the figures is low, making them hard to read.
Ans. Thank you for the observation. The figure has been zoomed out in the original layout for better clarity.
Q6. Section 3.8: Year-wise distribution of the bacterial isolates should also be provided.
Ans. Thank you for the suggestion. While we acknowledge the importance of year-wise distribution, the number of bacterial isolates collected per year at individual facilities was not sufficient for meaningful analysis or comparison. Therefore, we analysed cumulative data across the three-year period (2022–2024) to ensure robustness. The study presents the overall distribution of organisms isolated over this full timeframe from the selected facilities.
Q7. Section 3.9: In addition to comparative analysis with the ICMR-AMRSN Surveillance, the AST data with respect to the bacterial isolates collected in the study should be provided. This will provide an idea of the AMR status concerning each study facility, as well as whether there is a decline or increase in AMR during the study period.
Ans. Thank you for the suggestion. While we agree that year-wise facility-level AST trends can offer additional insights, the number of isolates per organism per year at each facility was limited, making such comparisons statistically unreliable. To address this, we have presented cumulative AST data across the three-year period, allowing for a more stable and representative analysis of AMR patterns at each study facility. Comparative analysis with ICMR-AMRSN further contextualises these findings.